# Vaccination Against Amyloidogenic Aggregates in Pancreatic Islets Prevents Development of Type 2 Diabetes Mellitus

**DOI:** 10.3390/vaccines8010116

**Published:** 2020-03-02

**Authors:** Elisa S. Roesti, Christina N. Boyle, Daniel T. Zeman, Marcos Sande-Melon, Federico Storni, Gustavo Cabral-Miranda, Alexander Knuth, Thomas A. Lutz, Monique Vogel, Martin F. Bachmann

**Affiliations:** 1Department of Rheumatology, Immunology and Allergology (RIA), University Hospital, University of Bern, 3010 Bern, Switzerland; federico.storni@insel.ch (F.S.); gcabral.miranda@gmail.com (G.C.-M.); monique.vogel@dbmr.unibe.ch (M.V.); martin.bachmann@me.ch (M.F.B.); 2Department for BioMedical Research (DBMR), University of Bern, 3010 Bern, Switzerland; 3Institute of Veterinary Physiology, University of Zürich, 8006 Zürich, Switzerlandtomlutz@vetphys.uzh.ch (T.A.L.); 4Department of Endocrinology, Diabetes, and Metabolism, University Hospital Basel, 4031 Basel, Switzerland; daniel.zeman@unibas.ch; 5Institute of Anatomy, University of Bern, 3010 Bern, Switzerland; msandemelon@yahoo.com; 6Spanish National Centre for Cardiovascular Research (CNIC), 28029 Madrid, Spain; 7Department of Visceral Surgery and Medicine, University Hospital of Bern, 3010 Bern, Switzerland; 8National Center for Cancer Care & Research (NCCCR), 3050 Doha, State of Qatar; kknuth@hamad.qa; 9Nuffield Department of Medicine, Centre for Cellular and Molecular Physiology (CCMP), The Jenner Institute, University of Oxford, Oxford OX3 7DQ, UK

**Keywords:** islet amyloid polypeptide, amyloid, T2DM, virus-like particle, vaccine

## Abstract

Type 2 diabetes mellitus (T2DM) is a chronic progressive disease characterized by insulin resistance and insufficient insulin secretion to maintain normoglycemia. The majority of T2DM patients bear amyloid deposits mainly composed of islet amyloid polypeptide (IAPP) in their pancreatic islets. These—originally β-cell secretory products—extracellular aggregates are cytotoxic for insulin-producing β-cells and are associated with β-cell loss and inflammation in T2DM advanced stages. Due to the absence of T2DM preventive medicaments and the presence of only symptomatic drugs acting towards increasing hormone secretion and action, we aimed at establishing a novel disease-modifying therapy targeting the cytotoxic IAPP deposits in order to prevent the development of T2DM. We generated a vaccine based on virus-like particles (VLPs), devoid of genomic material, coupled to IAPP peptides inducing specific antibodies against aggregated, but not monomeric IAPP. Using a mouse model of islet amyloidosis, we demonstrate in vivo that our vaccine induced a potent antibody response against aggregated, but not soluble IAPP, strikingly preventing IAPP depositions, delaying onset of hyperglycemia and the induction of the associated pro-inflammatory cytokine Interleukin 1β (IL-1β). We offer the first cost-effective and safe disease-modifying approach targeting islet dysfunction in T2DM, preventing pathogenic aggregates without disturbing physiological IAPP function.

## 1. Introduction

According to the International Diabetes Federation (IDF, 8th Edition 2017), approximately 425 million people worldwide are suffering from diabetes, and type 2 diabetes mellitus (T2DM) accounts for approximately 90% of all cases. Further, a 48% increase in patient numbers with T2DM is estimated to occur by 2045 (reaching > 600 million) [1,2]. T2DM is a chronic progressive disease characterized by obesity, insulin resistance and β-cell failure [2]. The onset of T2DM is mainly determined by a failure of the insulin-producing β-cells to secrete adequate levels of metabolically active insulin to maintain normoglycemia [3,4]. One hallmark of the disease is the occurrence of amyloid deposits in pancreatic islets [5,6], which is associated with β-cell apoptosis [7] and reduced β-cell mass [4,8]. The major constituent of amyloid deposits is the β-cell secretory product islet amyloid polypeptide (IAPP; also known as amylin). IAPP is co-secreted with insulin in a ratio of approximately 1:100 [9,10] and under physiological conditions it is present in a soluble, monomeric form and is responsible for the induction of central satiety, controlling gastric emptying, glucose homeostasis and for the suppression of glucagon release [11]. However, under certain conditions, IAPP starts to aggregate and to form cytotoxic intermediates, finally resulting in amyloid deposits. Among other mechanisms, such as increased IAPP secretion during prediabetes [12] or insufficient processing of IAPP as found in failing human islet grafts [13], IAPP is also one of the strongest activators of the NLRP3 inflammasome [14], which leads to activation and secretion of mature Interleukin 1β (IL-1β). Moreover, IAPP induces chemokine release from β-cells [15]. Among other factors, glucose and fatty acids, which are elevated in T2DM, induce IAPP and IL-1β transcription and secretion in human islets [16,17]. All of this leads to a vicious cycle which is governed by a glucose–IAPP–inflammasome–IL-1β axis. Interventions directed against the major pro-inflammatory players have shown some success in mouse models and clinical trials [15,18,19], but to date there is no disease-modifying drug available that directly targets islet amyloid and its associated complications. Currently, patients with T2DM are treated symptomatically with various drugs, including exogenous insulin itself, insulin sensitizers, GLP-1 analogues and DPP4 inhibitors that increase insulin secretion, as well as SGL2 inhibitors, which promote glycosuria. However, since IAPP and insulin are co-secreted from β-cells [9,20], anti-diabetic drugs that act as insulin secretagogues might also promote amyloid deposition.

Interestingly, species-specific IAPP amino acid sequences dictate the formation of pathogenic aggregates [20], where proline (Pro) residues present in the rodent but not in the human sequence, play a major role. Indeed, they act as β-sheet breakers preventing amyloidogenicity, and as such, in contrast to human IAPP, rodent IAPP is unable to aggregate and is not cytotoxic [21]. Therefore, mice that transgenically express human IAPP are widely used to study islet amyloidosis and its associated complications. Currently, an FDA-approved analogue of human IAPP (containing several amino acid substitutions that prevent it from aggregation) named Pramlintide is prescribed for mealtime usage for patients affected from diabetes under insulin prescription in order to fulfil IAPP missing functions [22]. However, this drug, together with insulin, even if disease modifying, does not prevent the onset of T2DM. For the development of a new disease-modifying therapy, we used virus-like particles (VLPs) [23], which are multi-subunit coat proteins (in our case derived from the bacteriophage Qβ) that self-assemble during their production in *Escherichia coli*. They are devoid of viral genomic material, therefore cannot replicate and are consequently safe to use as a carrier [24,25,26]. During their assembly process, they package negatively charged RNA [27] from their host machinery, *E. coli*, which is a Toll-like Receptor 7 (TLR7) ligand, acting as a natural adjuvant. Because of their optimal size (20 nm), they can drain free in the lymph nodes or be taken up from antigen-presenting cells and transported into the lymph nodes, and together with their repetitively structure, they can induce a strong antibody and memory response. Thus, VLPs combine great immunogenicity with good safety and tolerability. Up-to-date VLP-based vaccines already exist on the market, such as against human papilloma virus and against hepatitis B [28], and more are in clinical trials [29,30]. We have previously proven that VLPs can also be adopted as platform against amyloidogenic aggregates [31], chronic [32,33,34], allergic [35,36] and infection diseases [26,37,38]. In addition to humans, VLPs were successfully used as vaccines in companion animals [35,39,40]. In this preclinical study, we took advantage of VLP immunogenic properties and aimed at reducing amyloid deposits and thus preventing the progression to T2DM, therefore avoiding the accompanying secondary complications and life-threatening conditions.

Vaccination of IAPP transgenic mice against amyloidogenic aggregates of IAPP prevented the formation of IAPP aggregates in pancreatic islets in vivo, delayed the onset of amyloid-induced hyperglycemia and reduced local inflammation while restoring insulin production. Importantly, vaccination only targeted pathological amyloid aggregates and did not interfere with the physiological satiety-inducing effect of monomeric IAPP. Thus, vaccination against cytotoxic species of amyloid deposits might be a novel strategy to prevent T2DM and its associated complications in humans and companion animals [17,18,19,20,41].

## 2. Online Methods

### 2.1. Mice

C57BL/6JRccHsd mice were purchased (Envigo, The Netherlands) at the age of 6 weeks. FVB/N-Tg(Ins2-IAPP) RHFSoel/J mice [42] were purchased (Jackson Laboratory, Bar Harbor, Maine, MA, USA) at the age of 6 weeks and bred in our facility, maintained as a homozygous colony and on standard chow diet (diet # 3430; Granovit AG-Kliba Nafag, Switzerland). All animals were acclimatized to the facility for at least one week before experiments were performed. Experiments were approved by the animal ethic committee of the Swiss Cantonal Veterinary Office (Permission nr: BE70/18).

### 2.2. Pancreatic Human Tissue

Biopsies of patients’ pancreatic tissues with severe type 2 diabetes mellitus (under insulin treatment) were obtained from the Universitätsklinik für Viszerale Chirurgie und Medizin, Inselspital, Bern, Switzerland, with the approval of the Swiss Ethics Committees on research involving humans (project ID number: 2019-00068).

### 2.3. Production of VLPs Based Vaccines

VLPs obtained from the coat protein of the bacteriophage Qβ were expressed in *E. coli* strain JM109 with the expression vector pQβ10 and purified as previously described [41]. The N-terminal IAPP peptide without disulfide bond (s-s) (H-KCNTATCAT-OH), the C-terminal peptide (H-CGGTNVGSNTY-OH) and the C-terminal control peptide (H-CGGREPLNYLPL-OH) were purchased from Pepscan (The Netherland) and chemically coupled to VLPs via the heterobifunctional SMPH crosslinker (ThermoFisherScientific, USA). First, VLPs were derivatized to 7.5x molar excess of SMPH in PBS (pH 7.2) for 30 min at 25 °C on a shaking platform and after that, the non-bound crosslinker was removed using ZebaSpin columns (exclusion limit: 7 kDa, ThermoFisherScientific, USA). Secondly, the peptide was coupled with a 5-fold molar excess for 3 h at 25 °C in PBS on a shaking platform. The N-terminal peptide with disulfide bond (s-s) (H-KCNTATCATGGK[Aoa]-NH2, Pepscan, The Netherland) was coupled with a s4FB crosslinker (Solulink, US) in order to maintain the S-S bridge. First the VLPs were exchanged with a modification buffer (100 mM sodium phosphate; 150 mM NaCl, pH 8) and derivatized with 7.5 molar excess of s4FB. After removal of the unbound crosslinker, the buffer was exchanged with a conjugation buffer (100 mM citric acid, pH 6) to couple the peptide in a 5-fold molar excess. All vaccines were finally dialyzed against PBS (pH 7.2) and coupling efficiency was analyzed with 15% SDS-PAGE under denaturing conditions (gel loaded with ProSieve QuadColor marker (Lonza, Switzerland)). Densitometry analysis were performed with Fiji Image J using the gel quantification option.

### 2.4. Vaccines Immunogenicity

Female C57BL/6 mice were immunized subcutaneously at 6 weeks of age with 10 μg of either the differently conjugated vaccines or with non-conjugated VLPs as a negative control (150 μL on one ventral side) without any adjuvant addition. Mice received a boost injection at day 14 and 28. At each time point, blood was collected and serum was isolated using Microtainer Tubes (BD Bioscience, US). To measure the antibody response and to check which epitope the generated polyclonal antibody recognized, ELISA plates were first coated either with synthetic human IAPP (hIAPP; Bachem, Switzerland), rat IAPP (rIAPP; Bachem, Switzerland) or the corresponding peptide (coupled to ribonuclease (RNAse) A from bovine pancreas (Sigma-Aldrich, USA)) used for vaccination. Blocking was performed with 0.15% PBS-Casein for 2 h at room temperature, and serum was diluted 1:10 followed by a 1:3 serial dilution. To detect anti-mouse IgGs, a secondary anti-mouse IgG directly conjugated with horseradish peroxidase (Jackson ImmunoResearch Laboratories, USA) was used. A developing solution consisting of 0.01% TMB (3,3′,5′,5′-tetramethylbenzidine), 0.14% H_2_O_2_ and 30 mM citric buffer was added to the plate. After 2 min, the reaction was stopped with an equal volume of 1 M H_2_SO_4_ solution and OD at wavelength 450 was measured with an ELISA reader (BioTek, USA). OD50 values were defined as the reciprocal 50% dilution respectively to the maximal OD at 450. The best candidate vaccine (N-terminal with intact disulfide bond also called Qβ-Nterm (s-s) vaccine) was then further investigated in vivo.

### 2.5. Polyclonal IgG Antibody Purification

Pooled batches of serum obtained from Qβ-Nterm (s-s)-immunized mice were purified with a self-assembled protein G Sepharose 4Fast Flow (GE Healthcare Life Sciences, US) into disposable propylene columns (Qiagen, Germany). First, a 20 mM sodium phosphate (pH 7.4) binding buffer was passed through the column for equilibration, followed by sera loading and washing with binding buffer. Finally, a 0.1 M glycine-HCl, (pH 2.8) elution buffer was applied in order to elute the purified IgG. A final dialysis against PBS (pH 7.4) was performed.

### 2.6. hIAPP Monomer and fiBrils Preparation

For monomer preparation, lyophilized peptide was freshly dissolved in hexafluoroisopropanol (HFIP), filtered in Millex-GV 0.22 μm PVDF filters (MerckMillipore, USA) and lyophilized overnight. Prior to the experiment, hIAPP was dissolved in ddH_2_O. To generate fibrils, the same hIAPP dissolved in ddH_2_O was incubated at 37 °C overnight. To assess the presence of aggregates, a small amount was mixed with 60 μM Thioflavin T and checked under the fluorescent microscope Axio Imager.A2, and a Carl Zeiss AxioCam.

### 2.7. Dot Blot

Nitrocellulose membranes (Millipore Corporation, US) were plotted with 3 μL of 1 mg/mL of either rIAPP, hIAPP monomer, hIAPP fibrils or a control amyloidogenic protein amyloid beta (Aβ; Bachem, Switzerland). After drying, the membrane was boiled at 900 W for 5 min in a 4.3 mM Na_2_HPO_4_ 1.4mM KH_2_PO_4_ containing buffer to expose epitopes and increase antigen binding; washed and blocked with 50 mM Tris-buffer containing 0.1% Tween 20 and 2.5% casein. Overnight incubation with the purified polyclonal antibody was performed and finally a secondary anti-mouse IgG directly conjugated with horseradish peroxidase (Jackson Immuno Research Laboratories, Baltimore Pike, West Grove, PA, USA) was used. Development of the membrane was performed using a SuperSignal^TM^ West Pico PLUS Chemiluminescent Substrate (Thermo Fisher Scientific, Federal Way, WA, USA) and pictures were acquired with an Azure C300 Imaging System (Axon Lab AG, Baden, Switzerland). Fiji Image J was used to quantify the intensity of the dots. A background subtraction of 25 pixel was applied before quantification of the intensity od density. Monomeric IAPP and IAPP/Aβ fibrils were assessed with Thioflavin as described above.

### 2.8. Assessment of IAPP-Induced Satiety Following Refeeding

C57BL/6JRccHsd male mice were immunized at 6 weeks of age and boosted at week 8 and 10 as described above. IgG titers were confirmed measuring collected sera (day 14 and day 34) as described above. On day 24, mice were single-housed in cages with external hoppers. On days 36, 38, 41 and 43, mice were fasted during the light phase for a period of 12 h. Immediately prior to dark onset, 20, 100 or 500 μg/kg IAPP or saline was randomly administered and food intake was measured 1, 2, 4 and 22 h after treatment. Baseline correction for each time point was calculated based on a 100% intake for the control saline group.

### 2.9. Monitoring and Immunization of the Transgenic Mice

After weaning at the age of 4 weeks, male homozygous hIAPP transgenic mice received high-fat diet (45% kcal from fat; diet # 2126; Granovit AG-Kliba Nafag, Kaiseraugst, Switzerland). At 6 weeks of age (day 0), mice received 10 μg of Qβ-N-terminal (s-s) vaccine and were boosted at week 8 (d14) and 10 (d28). Body weight and 14-h fasting glycemia (Accu-Check Aviva (Roche, Basel, Switzerland)) were measured weekly. Serum was obtained and antibody titer against hIAPP, rIAPP and N-terminal (s-s) peptide was measured to ensure efficacy and specificity of the vaccine, as explained above.

### 2.10. Immunofluorescence of Mouse and Human Tissue

Mice were sacrificed and perfused with 4%PFA, pancreata were further fixed for 4 h in 4% PFA, gradually changed to 70% (*v*/*v*) ethanol and processed in a STP 120 Spin Tissue Processor l Myr according to the manufacturing instructions (Thermo Fisher Scientific, Federal Way, WA, USA) for paraffin embedding. Prior to all staining, tissues were deparaffinised (xylol, 100% (*v*/*v*) ethanol, 94% (*v*/*v*) ethanol, 70% (*v*/*v*) ethanol, 50% (*v*/*v*) ethanol, ddH_2_O) and a 15 min antigen retrieval (except for the Thioflavin S staining) was performed with Tris-EDTA buffer pH 9.0 (10mM Tris Base, 1mM EDTA, 0.05% Tween 20) in a 900W-microwave. The 1% Thioflavin S (T1892, Sigma-Aldrich, Port Washington, WI, USA) in ddH_2_O was used to stain for amyloid aggregates directly after deparaffinisation. For the detection of β-cells, a rabbit monoclonal [EPR17359] IgG to insulin (ab181547, Abcam, Cell Signaling Technology Inc, Cambridge, UK) followed by a goat anti-rabbit IgG conjugated to biotin (Nordic-MUbio, Susteren, The Netherlands) and a streptavidin conjugated Alexa546 (s11225, Molecular Probes, Carlsbad, CA, USA) were used. Monoclonal anti-IL-1β (F-5, sc-515598, Santa Cruz) followed by a Cy5-conjugated anti-mouse IgG (Jackson ImmunoResearch, Baltimore Pike, West Grove, PA, USA) was used to identify inflamed cells. For intracellular staining, sections were permeabilized with 0.5%-TritonX-100 in PBS. As a blocking solution bovine serum (3%), casein (0.5%) and NaN_3_ (0.1%) was prepared, while when performing intracellular staining the blocking buffer was made up of BSA (2%) and 0.5%-Triton X-100. In the presence of biotinylated antibodies an avidin/biotin blocking kit (Vector Laboratories, USA) was used to block endogenous biotin according to the manufacturing instructions. All pictures were acquired with an Axio Imager. A2 and a Carl Zeiss AxioCam. Tomographic pictures were acquired, after Thioflavin S staining only as described above, with a 3D Cell Explorer-fluo (Nanolive, Lausanne, Tolochenaz, Switzerland).

### 2.11. Immunofluorescence Quantification

For all the quantifications, Fiji Image J (NIH, Bethesda, MD, USA) was used. For the amyloidogenic aggregates, the area occupied by the aggregates was selected by composite selection and divided by the total islet area. For the insulin and IL-1β quantification, single cells were manually counted using the cell counter option and divided by the total number of nuclei. For the insulin correlation, the mean insulin intensity of each islet was measured and plotted against the corresponding total area of the islet.

### 2.12. Statistics

Statistical analyses were performed within two groups with Mann-Whitney test assuming non-parametric distribution; 2-way ANOVA with prior normality test; and Sidak’s multiple comparison when appropriated with GraphPad PRISM 6.0 (Graph-Pad Software, Inc., La Jolla, CA, USA). Statistical significance is displayed as *p* ≤ 0.05 (*), *p* ≤ 0.01 (**), *p* ≤ 0.001 (***), *p* ≤ 0.0001 (****). The required number of mice to reach a power of 80% was calculated based on literature and preliminary experiments. For each experiment and group “n” is indicated in the figure legend.

## 3. Results

### 3.1. Qβ-VLPs Vaccines Against IAPP Aggregates Generation and Analysis

Human IAPP assembles into oligomers and fibrils, thereby forming β-sheet stacks stabilized by hydrogen bonds [42] and supported by the IAPP core structure [43,44] consisting of an N-terminal peptide linked by a disulfide (s-s) bond followed by two parallel β-sheets and a final amidated C-terminal peptide. Because of their likely exposed epitopes, we designed vaccines displaying different IAPP-derived peptides in order to induce antibodies which selectively recognize oligomeric IAPP but do not bind and neutralize free, soluble IAPP. For this, various IAPP peptides were coupled to Qβ-VLPs and used to immunize mice in order to produce specific antibodies**.**

To limit affinity for monomeric bioactive IAPP, we used short peptides which usually induce low affinity antibodies [31] but may be able to bind with high avidity to aggregated “multi-meric” immobilized IAPP. Two different N-terminal peptides, N-term (s-s) and N-term, with and without a disulphide bond (s-s) between cysteine 2 and 7, respectively, a C-terminal peptide (C-term) and a C-terminal peptide of pro-IAPP (C-term-Pro) were coupled to Qβ-VLPs (Figure 1a,b). For this, Qβ-VLPs were chemically derivatized to heterobifunctional cross-linkers via Lys-aa (SMPH for the N-term, C-term and C-term-Pro peptides and s4FB for the N-term(s-s) peptide) and secondly coupled to the mentioned peptides via Cys-aa or an Aox-group for the SMPH and the s4FB, respectively. The following vaccines were obtained: Qβ-Nterm (for coupling to N-term peptide without disulfide bond); Qβ-Nterm (s-s) (for coupling to the N-term (s-s) peptide with the disulfide bond); Qβ-Cterm, (for coupling to the C-term peptide); and Qβ-Cterm-Pro (for coupling to the C-term-Pro peptide). The coupling efficiency was analyzed by SDS-PAGE (Figure 1c) and coupling efficiency was quantified by densitometry (Figure 1d). All vaccines were successfully coupled but with varying efficiency. The Qβ-N-term vaccine showed a poor coupling with maximally one peptide per subunit (Figure 1c,d). In contrast, the Qβ-N-term (s-s) showed a very efficient coupling with up to 3 peptides per subunit covering approximately 87% of the surface Qβ-VLPs occupied (Figure 1c,d). The Qβ-C-term vaccine gave a similar result to the Qβ-N-term without the disulfide bond with relatively low coupling, while the C-term-Pro resulted in a good coupling efficiency (Figure 1c,d).

### 3.2. IAPP Peptides Coupled to Qβ-VLPs are Highly Immunogenic

To test whether the designed vaccines were able to mount an antibody immune response, 6-week-old C57BL/6JRccHsd mice were immunized with 10 μg of each vaccine and received booster injections two and four weeks later (Figure 2a). Serum was collected every two weeks (with a final collection at day 49) and the presence of specific antibodies was measured by ELISA assays using as coating antigens rodent IAPP (rIAPP), human IAPP (hIAPP) and peptides used for vaccination coupled to RNase (Figure 2b). The amino acid sequences of fully processed rodent and human IAPP only differ in three amino acids (Appendix A). These substitutions are found in the amyloidogenic sequence (amino acids 20–29), which is thought to be responsible for IAPP aggregation [43]. Indeed, the final quaternary structure consists of a β-sheet conformation for the amyloidogenic human IAPP, and an α-helical structure for the non-aggregating rodent IAPP. As human IAPP is prone to aggregation, this allowed us to test recognition of oligomeric (human) and monomeric (human and rat) IAPP, which both have the same identical N-terminus and C-terminus peptide sequences used for immunization, as displayed in Figure 1a,b. All polyclonal IgGs recognized their specific peptide coupled to RNase as carrier protein, with Qβ-N-term(s-s)-IgGs giving the highest titer (Figure 2c–f), confirming that all vaccines are immunogenic. Interestingly, hIAPP was only recognized by antibodies induced by the Qβ-N-term (s-s) vaccine, but not by the other candidates (Figure 2c,e,f). Noteworthy, lack of the disulfide bond in the otherwise identical Qβ-N-term vaccine completely prevented binding, suggesting that the conformation of the N-terminus may play an important role for recognition of human IAPP. Moreover, none of the vaccine induced antibodies recognizing rIAPP. These results indicate that Qβ-N-term (s-s) induces antibodies selectively recognizing aggregated IAPP. For this reason, further analyses were performed using the Qβ-N-term (s-s) vaccine.

### 3.3. Specific IgGs Recognize Aggregated, But Not Soluble Human IAPP

Next, we investigated whether Qβ-N-term (s-s) vaccination-induced antibodies discern between monomeric and oligomeric hIAPP. We found that it recognizes IAPP oligomers/fibrils, but not monomeric IAPP, rodent IAPP or a negative control composed of amyloidogenic Aβ peptide (Figure 3a). To prove that the correct molecular forms of IAPP were plotted on the nitrocellulose membrane, hIAPP monomer and fibrils were stained with Thioflavin T and imaged (Figure 3b). hIAPP and Aβ fibrils were readily detected, presenting aggregates sized up to 50 nm (Figure 3b). As expected, monomeric human IAPP and rodent IAPP were not forming or present in a fibril-form (Figure 3b). These results further support the specificity of the Qβ-N-term (s-s) vaccine-derived polyclonal antibodies and show their specificity for oligomeric/fibrillic but not soluble monomeric IAPP.

### 3.4. Qβ-N-Term (s-s) Vaccine-Derived IgGs Bind Specifically to IAPP Aggregates in Human Pancreatic Tissue

In order to test the recognition of human IAPP’s aggregates in human tissue, we tested IgGs derived from mice immunized with the Qβ-N-term (s-s) vaccine on a pancreatic tissue section from a patient suffering from severe T2DM, expected to show amyloid deposition. Antibodies specifically bound to the abundant amyloidogenic aggregates present in the pancreatic islets which were also stained by Thioflavin S (ThioS) (Figure 3c, upper panel). Compared to staining of hyperglycemic transgenic mice expressing hIAPP (Figure 3c, lower panel), polyclonal IgGs did not bind to control section of healthy human pancreatic tissue (Figure 3d, higher panel) and to normoglycemic transgenic mice (Figure 3d, lower panel).

### 3.5. IAPP Function Remains Intact After Immunization with the Qβ-N-Term (s-s) Vaccine

To test whether our vaccine impais the physiological function of the monomeric IAPP, we assessed IAPP-induced satiety in Qβ-N-term (s-s) or Qβ vaccinated wild-type mice. Mice were immunized and boosted 2 and 4 weeks later (Figure 4a) either with the Qβ-N-term (s-s) vaccine or with Qβ only. Antibody responses were analysed for recognition of the N-term (s-s) peptide coupled to RNase (RNase-N-term (s-s)), hIAPP or rIAPP (Figure 4b). In the presence of high levels of specific IgG (Figure 4b), mice were single caged, and on day 36, day 38, day 41 and day 43, fasted during the light phase for 12 h and injected immediately prior to dark onset with a single dose of either 20, 100 or 500 μg/kg synthetic rodent IAPP or saline. One hour after IAPP injection, control and vaccinated groups both showed a similar decreased food intake in a dose dependent way (Figure 4c); following 500 μg/kg rodent IAPP, the control and vaccinated groups reduced food intake to 60 and 50% of vehicle, respectively (main effect of Iapp, *p* = 0.05), with no significant difference between saline and IAPP 500 group at baseline correction (Figure 4d) in both Qβ only and Qβ-N-term (s-s)-immunized mice. This indicates that monomeric IAPP is still functional, confirming that monomeric IAPP remains undetectable from the polyclonal antibodies induced from the Qβ-N-term (s-s) vaccine. Cumulative food intake and time points with baseline corrections are shown in Appendix A.

### 3.6. Qβ-N-Term(s-s) Vaccination Ameliorates T2DM in hIAPP Transgenic Mice

To assess the potential efficacy of the vaccine, we tested the candidate Qβ-N-term (s-s) vaccine in an in vivo mouse model of islet amyloidosis expressing human IAPP under the control of the Rat Insulin II Promoter (RIP). Due to the hIAPP expression in the pancreatic β-cells, amyloid aggregates deposit in the pancreatic islets and therefore lead to gradual β-cell destruction. Thus, this model closely mimicks human T2DM, including islet amyloidosis. To drive insulin and IAPP secretion and thus increase amyloid deposition, we put the mice on a high-fat diet (HFD). Tomographic images confirm aggregates in the islets of hIAPP transgenic mice but not in islets of non-transgenic control mice (Figure 5a).

hIAPP transgenic mice were fed HFD starting at 4 weeks of age and received the first immunization at week 6 (day 0) with Qβ-N-term (s-s) or uncoupled Qβ as control. Mice were then boosted at week 8 (day 14) and at week 10 (day 28) (Figure 5b). Sera were collected weekly (last collection at day 35; Figure 5b) and analyzed for specificity (Figure 5c). IgGs derived from Qβ control-immunized mice, as expected, did not recognize any peptide complex (Figure 5d). In contrast, IgGs from immunized mice recognized the RNase-N-term (s-s) peptide complex as well as the hIAPP (Figure 5e). To detect the onset of T2DM and disease progression, fasting glycemia was measured weekly. The Qβ control mice showed progressive hyperglycemia starting at 9 weeks of age (Figure 5f). In contrast, the Qβ-N-term (s-s)-immunized mice only showed mild hyperglycemia starting at 12 weeks of age. Importantly, Qβ-N-term (s-s) vaccination also protected mice from hyperglycemia-induced loss in body weight (Figure 5g).

### 3.7. Qβ-N-Term (s-s) Vaccinated hIAPP Transgenic Mice Show Significantly Fewer IAPP Aggregates and IL-1β-Positive Cells

Studies demonstrated that amyloid deposits are present in more than 90% of post-mortem diabetic patients [45]. To assess whether our candidate vaccine was able to delay the onset of hyperglycemia due to decreased deposition of amyloid aggregates present in the pancreatic islets, we quantified the occupancy of amyloid deposits. To this end, paraffin-embedded pancreatic sections of 12-week-old immunized FVB/N-Tg(Ins2-IAPP) transgenic and control mice were stained for insulin and Thioflavin S. Interestingly, Qβ-N-term (s-s)-immunized mice showed significantly reduced relative area occupied by amyloid in comparison to Qβ injected mice that was similar to the wild-type control mice (Figure 6a,b), indicating that the vaccine is able to prevent amyloid deposition.

Next, we investigated the correlation between insulin production and islet area in the three experimental groups. Wild-type FVB control mice showed a negative correlation between insulin mean intensity and islet area, while Qβ-immunized control mice did not show this negative correlation. Interestingly, Qβ-N-term (s-s)-immunized mice showed a clear, albeit reduced negative correlation (Figure 6d). Further, loss of β-cell mass is a hallmark of human T2DM [8,46] and amyloid deposition is positively correlated with β-cell apoptosis in humans [7]. Thus, we next investigated whether hIAPP vaccination prevents amyloid-induced loss of β-cell mass. Qβ control mice indeed showed significantly less insulin-positive cells when compared to Qβ-N-term (s-s)-immunized mice, indicating that the vaccine has a protective effect for the insulin-producing β-cells (Figure 6e,f).

Finally, we investigated the mechanism of the protective effect of Qβ-N-term (s-s) vaccination. As IAPP is one of the strongest activators of the NLRP3 inflammasome^14^ and this complex activates the pro-inflammatory master cytokine IL-1β, we quantified IL-1β-positive β-cells in pancreas sections of 12-week-old hIAPP transgenic mice. Intriguingly, compared to control Qβ mice, Qβ-N-term (s-s)-immunized mice had reduced numbers of IL-1β-positive β-cells (Figure 6f). The rate of IL-1β-positive β-cells in immunized mice did not differ from wild-type control mice, suggesting that the vaccination not only reduces amyloid deposition but also prevents amyloid-induced upregulation of the pro-inflammatory cytokine IL-1β.

Collectively, our data indicate that Qβ-N-term (s-s) is efficient at delaying onset of hyperglycemia, caused by a reduction of amyloidogenic deposits together with a local reduction of the pro-inflammatory cytokine IL-1β.

## 4. Discussion

T2DM is a chronic disease affecting increasing numbers of people every year; in addition, the onset of the disease is occurring at an earlier age [2]. A hallmark of T2DM is the occurrence of IAPP-derived amyloid aggregates in the pancreatic islets, which is associated with β-cell toxicity and inflammasome-mediated IL-1β production and inflammation [34], leading to the characteristic loss of β-cell mass in type 2 diabetes mellitus [4]. Numerous efforts have been aimed at inhibiting amyloid toxicity by inhibiting IAPP aggregation or blocking downstream effects, but to date there is no successful in vivo therapy available that reduces amyloid deposition.

VLPs are used as vaccine platforms as they are safe but potent immunogenic nanoparticles. Displaying antigens on VLPs results in strong antibody responses as they carry a “viral fingerprint” [23]. Displaying the disulfide bond-linked N-terminus of hIAPP on VLPs resulted in a vaccine candidate that (1) induces strong antibody responses in the absence of adjuvants and (2) has high specificity for aggregated IAPP while the beneficial physiological function of monomeric IAPP is not affected.

Our data show that vaccination of amyloid-prone transgenic mice results in reduced amyloid deposits and lower levels of the master cytokine IL-1β within islets, while production of insulin was preserved and comparable to levels of control wild-type mice. Consequently, the onset of disease was significantly delayed. However, hyperglycemia was not completely prevented. This may be due to very high levels of IAPP in this mouse model, much higher than what is typically observed in humans. Consequently, hyperglycemia develops within weeks while in humans it normally takes decades to develop β-cell loss that manifests in overt hyperglycemia. Thus, in this mouse model of islet amyloidosis, intracellular systems including chaperones and proteases clearing mechanism may be congested [47,48], making it more difficult to prevent the progression of the disease. We would expect that the magnitude of our vaccination effect is bigger in more mild models that more closely resemble human type 2 diabetes mellitus. Vaccination of amyloid-prone mice also prevented hyperglycemia-induced loss in body weight, further showing that downstream pathological consequences of islet amyloidosis can be prevented by our vaccination.

There is no dominant opinion about the exact localization of the amyloid deposits and both intracellular and extracellular IAPP aggregates have been described [49]. As our antibodies primarily recognize and remove extracellular aggregates, our data suggest that a large fraction of the aggregates is localized in the extracellular space and that removal of these extracellular aggregates alone is sufficient to improve symptoms of T2DM, including a decrease in glucose level and reduced production of the pro-inflammatory cytokine IL-1β.

Interestingly, the disulfide bond in the N-terminus was essential to induce antibodies that recognize IAPP aggregates with the desired specificity, while a linear N-terminus or the C-terminus failed to do so. This is consistent with the notion that amyloidogenic deposits in the islets and monomeric IAPP present a structure with an exposed “folded” N-terminus as it is the case for the N-terminal(S-S) peptide, indicating that this peptide represents a preferred target structure for inducing anti-human IAPP antibody [50]. Further, it is still under debate whether prefibrillar IAPP structures, fibrils or amyloid deposits are the cytotoxic species [51]. Our data show that removal of aggregated amyloid deposits ameliorates amyloid-induced hyperglycemia, suggesting that end-stage plaques might not be as inert as postulated and that targeting these species is a valid strategy to counteract amyloid-induced β-cell damage.

It is important to appreciate that IAPP is not only involved in pathological amyloid aggregation but has a potent satiety function [52]. Indeed, IAPP is the second most abundant β-cell secretory peptide. It is co-secreted in a molar ratio of 1:100 with insulin and is a strong inducer of satiety via central pathways [53]. It is therefore important to keep the physiological function of IAPP in mind when designing targets to counteract islet amyloidosis. Our vaccination has that advantage that it only targets aggregated but not soluble IAPP. This may be explained by a simple avidity effect, as we have previously shown for alpha-synuclein, which causes Parkinson’s disease by protein aggregation. In fact, alpha-synuclein is recognized with very low affinity in contrast to aggregates which are bound with high avidity by antibodies induced with a peptide based vaccine [31]. Thus, monovalent binding of the peptide results in low affinity, but bivalent binding of the aggregates results in high avidity. Importantly, we also show on a functional level that the beneficial satiety-inducing activity of IAPP was preserved in immunized animals, confirming an absence or low affinity towards monomeric IAPP.

Other groups have also developed vaccinations against amyloidogenic peptides [54,55]. Lin et al. vaccinated IAPP-transgenic mice with Alzheimer’s Aβ oligomers which share some similarity with IAPP aggregates. Although high titers of anti-IAPP antibodies were obtained, hyperglycemia and loss of β-cell mass were not prevented. In fact, β-cell apoptosis was even increased, suggesting that Aβ oligomers are not a suitable antigen to target islet amyloidosis. Bram et al. [55] obtained an antibody response against IAPP by injecting toxic IAPP oligomers. While this approach normally leads to an amyloid-seeding effect and thus increased amyloid deposition, β-cell apoptosis and concurrent hyperglycemia [56], they co-administered complete Freund’s adjuvant which seemed to prevent hyperglycemia and preserve pancreatic insulin content but to reduce body weight. Islet IAPP levels were decreased 3-fold while islet amyloid deposition was not investigated. While preventing hyperglycemia is favorable, reducing IAPP might abolish IAPP’s physiological role to induce satiety, a feature that was not assessed. Further, reduced body weight in oligomer-treated mice (despite normalized glycemia) might point to a pathological off-target effect. Our data suggest that targeting IAPP aggregates and not pre-fibrillary structures has advantages over the approach by Bram et al. [55], mainly by not tempering with the physiological satiety role of pre-fibrillary IAPP and by increased safety due to not requiring co-administration of adjuvants and by not risking to boost amyloid burden by inducing amyloid seeding.

In conclusion, we demonstrated that vaccination against IAPP has potential to prevent amyloid-mediated complications by inducing antibodies selectively recognizing amyloid deposits and oligomers while leaving the physiological satiety function of the hormone intact. Thus, our VLP-based IAPP vaccine may represent a breakthrough therapy for human T2DM and other species affected by this disease.

## Figures and Tables

**Figure 1 vaccines-08-00116-f001:**
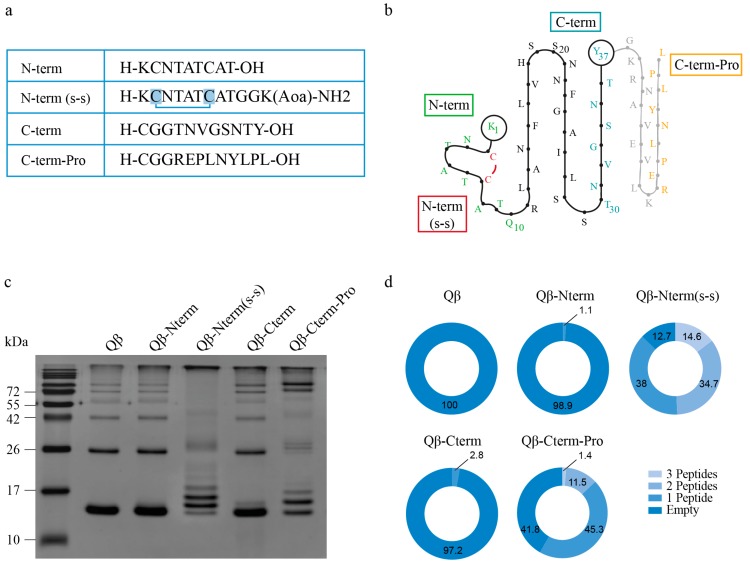
Qβ-VLPs based vaccine generation against candidate IAPP peptides. (**a**,**b**) Human IAPP amino acids (aa) sequence of the chosen peptides being coupled in a linear (**a**) and secondary (**b**) structure view: A N-terminal peptide without disulfide bond (N-term, in green); a N-terminal peptide with disulfide bond (s-s) between cystein 2 and 7 (N-term(s-s), in green and in red the disulphide bond)); a C-terminal peptide presents in the final 37-aa sequence (C-term, in light blue); and a C-terminal peptide (C-term-Pro, in orange) presents in the aa-sequence before final cleavage to the 37-aa long IAPP. (**c**) Firstly, Qβ-VLPs were chemically derivatized to heterobifunctional cross-linkers via Lys-aa (SMPH for the N-term, C-term and C-term-Pro peptides and s4FB for the N-term (s-s) peptide) and secondly coupled to the mentioned peptides via Cys-aa or an Aoa-group for the SMPH and the s4FB, respectively. The following vaccines were obtained: Qβ-Nterm (for coupling to N-term peptide without disulfide bond); Qβ-Nterm (s-s) (for coupling to the N-term (s-s) peptide with the disulfide bond); Qβ-Cterm, (for coupling to the C-term peptide); and Qβ-Cterm-Pro (for coupling to the C-term-Pro peptide). Coupling efficiency was analysed on a 15% SDS-PAGE under reducing condition. (**d**) To have detailed information of how many peptides per subunits were present, densitometry analysis was performed using Fiji ImageJ. The percent of the measured bands is displayed. Total area is represented as 100%. Qβ-VLPs, Virus-like particles derived from the bacteriophage Qβ; (s-s), disulphide bond; aa, amino acid; Lys, Lysin; Cys, Cystein, Aoa, aminooxyacetic group on the side chain of Lysin.

**Figure 2 vaccines-08-00116-f002:**
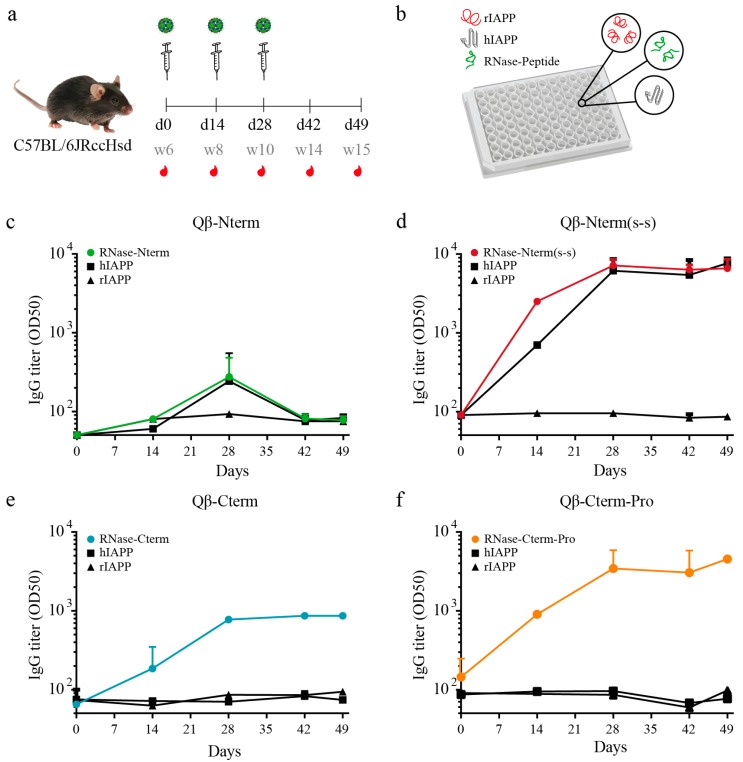
Immunogenicity control in C57BL/6 mice with the generated candidate vaccines. (**a**) Immunization scheme of 6 weeks old C57BL/6 mice. Mice were immunized at day 0 (d0), corresponding to the age of 6 weeks (w6). Boost was performed at day 14 (d14, w8) and day 28 (d28, w10). Blood was collected at day 0, day 14, day 28, day 42 and day 49 and serum was isolated. (**b**) 96-well ELISA plates were coated either with the corresponding peptide coupled on the vaccine used for immunization (in this case coupled to RNase), or with rat IAPP (rIAPP) or human IAPP (hIAPP). (**c**–**f**) Total serum IgG of the mice receiving: the Qβ-Nterm vaccine (*n* = 4) (**c**) recognizing the RNase-Nterm peptide (in green); the Qβ-Nterm (s-s) vaccine (*n* = 4) recognizing the RNase-Nterm (s-s) peptide (in red); the Qβ-Cterm vaccine (*n* = 4) recognizing the RNase-Cterm peptide, and the Qβ-Cterm-Pro vaccine (*n* = 4) recognizing the RNase-Cterm-Pro peptide (in orange). Total IgG recognizing the hIAPP (square) and rIAPP (triangle) are shown in black. OD50 values are calculated as the corresponding dilution reaching to the half value of OD450. Data are the means ± SEM. d, day; w, week; rIAPP, rat IAPP; hIAPP, human IAPP; RNase-peptide, peptide covalently coupled to the corresponding peptide being either Nterm, Nterm (s-s), Cterm or Cterm-Pro.

**Figure 3 vaccines-08-00116-f003:**
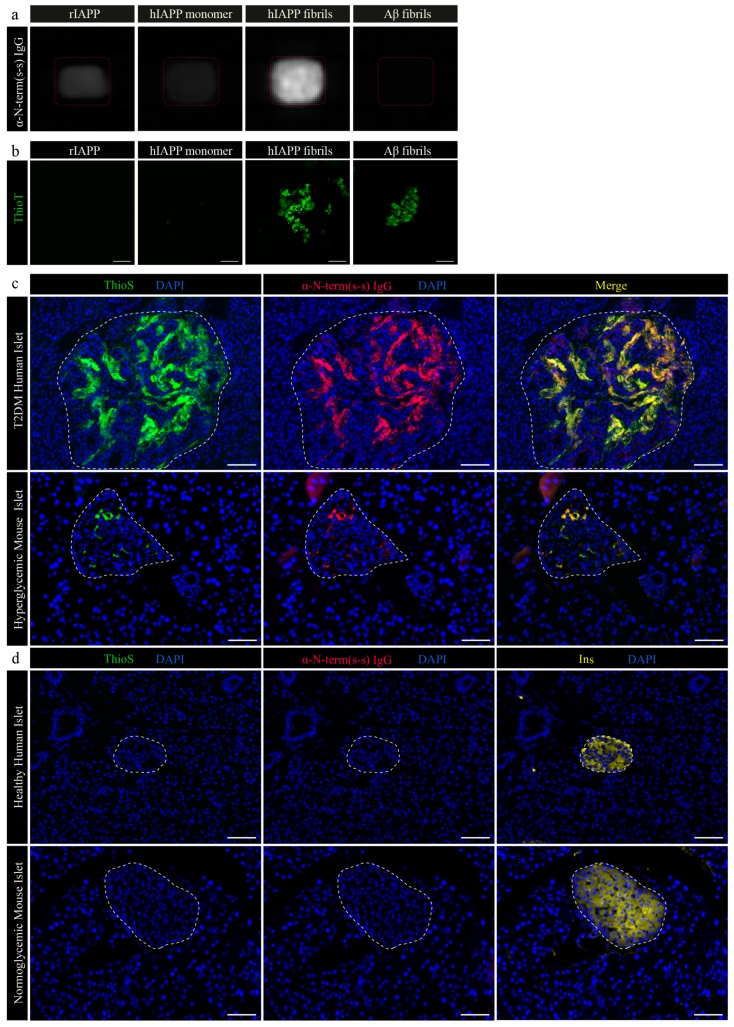
IgGs derived from C57BL/6 mice immunized with the Qβ-N-term (s-s) vaccine recognize only hIAPP aggregates on human and on hyperglycemic hIAPP transgenic mouse pancreatic tissue. (**a**) Nitrocellulose membrane coated either with rat IAPP (rIAPP), human IAPP monomers (hIAPP monomer), human IAPP fibrils (hIAPP fibrils) or a control amyloidogenic protein forming aggregates derived from amyloid beta (Aβ fibrils) followed incubation with purified polyclonal antibodies. After incubation with an anti-IgG conjugated to horseradish peroxidase, the membrane was developed with a chemiluminescent substrate. (**b**) Immediately after membrane blotting with the specified antigens, solutions were stained with a ThioflavinT (ThioT) solution and imaged to prove the state of the coated proteins. (**c**) Representative human pancreatic islet from a severe type 2 diabetes mellitus (T2DM) patients and a hyperglycemic transgenic mouse expressing hIAPP. Tissues were stained with ThioflavinS (ThioS) to confirm the presence of amyloidogenic aggregates (**c**, left panel, ThioS in green, DAPI in blue) and with an IgG derived from mice immunized with the Qβ-N-term(s-s) vaccine (**c**, middle panel, α-N-term (s-s) IgG in red, DAPI in blue). A co-localization staining (right panel, merge) was generated to authenticate the specificity of the IgGs. (**d**) Representative human pancreatic islet from a healthy pancreas and a wild-type mouse. ThioS staining (left panel) and polyclonal staining were checked on healthy human pancreatic tissue and wild-type normoglycemic mouse tissue. β-cells were visualized by staining with an anti-insulin (Ins, right panel) antibody. rIAPP, rat IAPP; hIAPP, human IAPP; Aβ, amyloid beta; α-N-term (s-s) IgG, polyclonal IgG derived from the Qβ-N-term (s-s) vaccinated C57BL/6 mice; ThioT, ThioflavinT; T2DM, type 2 diabetes mellitus; ThioS, ThioflavinS. Scale bars: 20 μm.

**Figure 4 vaccines-08-00116-f004:**
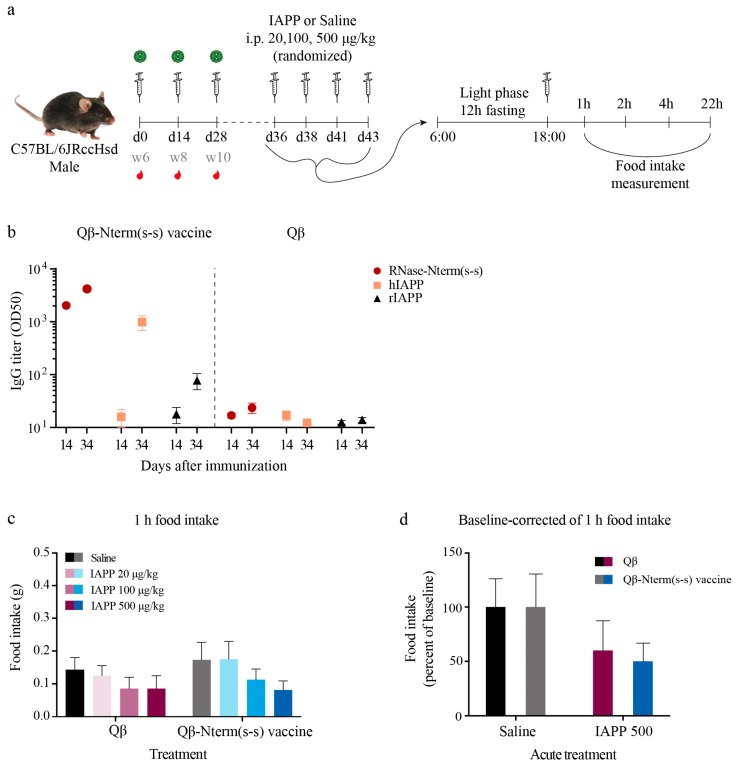
IAPP physiological function is maintained in the presence of anti-IAPP antibodies. (**a**) C57BL/6 male mice were immunized with the Qβ-Nterm (s-s) vaccine (*n* = 16) or with the uncoupled Qβ-VLPs (*n* = 14) at day 0 (week 6), day 14 (week 8) and day 28 (week 10), serum was collected at day 14 and day 34 (**b**) to check antibody titer. On days 36, 38, 41 and 43, mice were fasted during the light phase period for 12 h. Mice then received either IAPP or the control saline solution immediately prior to dark onset and food intake was measured 1, 2, 4 and 22 h after injection. (**b**) Serum antibodies were analyzed for recognition of the N-term (s-s) peptide coupled to RNase (RNase-N-term(s-s) in red), hIAPP (in peach) and rIAPP (in black) for the Qβ-Nterm (s-s) (left panel) and the Qβ-VLP (right panel)-immunized mice. (**c**) Cumulative food intake after 1-h refeeding and (d) baseline correction for the IAPP versus saline group in Qβ and Qβ-Nterm (s-s)-immunized mice. Additional time points can be seen in Appendix A. Data are the means ± SEM. RNase-N-term (s-s), N-term (s-s) peptide coupled to RNase; rIAPP, rat IAPP; hIAPP, human IAPP. Statistical test in (**c**,**d**): 2-way ANOVA and Sidak’s multiple comparison. No significant difference was observed in (**c**,**d**) (*p* > 0.05).

**Figure 5 vaccines-08-00116-f005:**
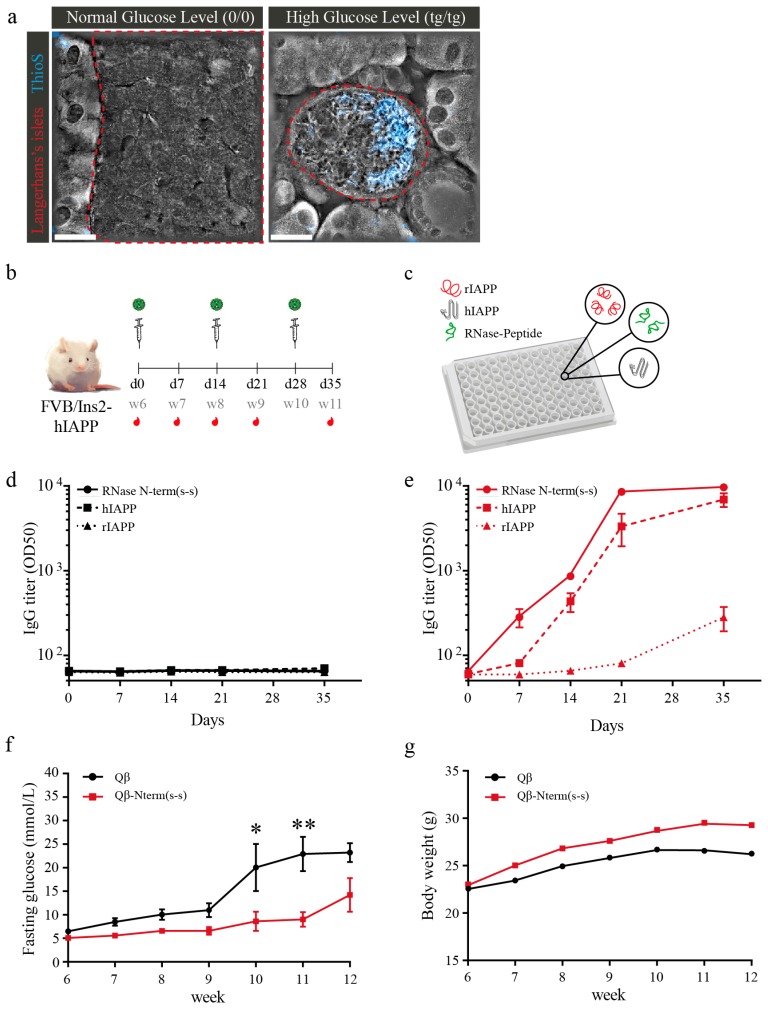
Immunization with the Qβ-Nterm(s-s) vaccine protects hIAPP transgenic mice from hyperglycemia and body weight loss. (**a**) Representative tomographic pictures merged with Thioflavin S (in blue) staining of pancreatic section of wild-type FVB and homozygous hyperglycemic male mice. Pancreatic islets are outlined in red. (**b**) Vaccination schedule of homozygous male FVB/Ins-hIAPP transgenic mice receiving either Qβ-Nterm (s-s) vaccine (*n* = 8) or uncoupled Qβ control (*n* = 7). (**c**) ELISA plates were coated either with the RNase-Nterm (s-s) peptide complex, with human IAPP (hIAPP), or with rat IAPP (rIAPP) to check for specific epitope recognition. (**d**) Total IgG titer of mice receiving the Qβ control. IgGs did not give positive signal for recognition of the RNase-Nterm (s-s) peptide complex (continuous black line), the hIAPP (dash black line) and the rIAPP (pointed black line). (**e**) Total IgG titer of mice receiving the Qβ-Nterm (s-s) vaccine. IgGs derived from the immunized mice recognized the RNase-Nterm (s-s) peptide complex (continuous red line), the hIAPP (dash red line) and the rIAPP (pointed red line). Qβ-immunized mice IgG did not give a positive signal for recognition of the RNase-Nterm (s-s) peptide complex (continuous black line), the hIAPP (dash black line) and the rIAPP (pointed black line). (**f**) Overnight fasting glycemia level and (**g**) body weight development in Qβ-Nterm (s-s) immunized (in red) and Qβ control group (in black). Statistical test; two-way ANOVA. Data are the means ± SEM. * *p* < 0.05, ** *p* < 0.01. ThioS, ThioflavS; (0/0), wildtype; (tg/tg), homozygous; RNase-N-term (s-s), N-term (s-s) peptide coupled to RNase; rIAPP, rat IAPP; hIAPP, human IAPP. Scale bars: 20 μm.

**Figure 6 vaccines-08-00116-f006:**
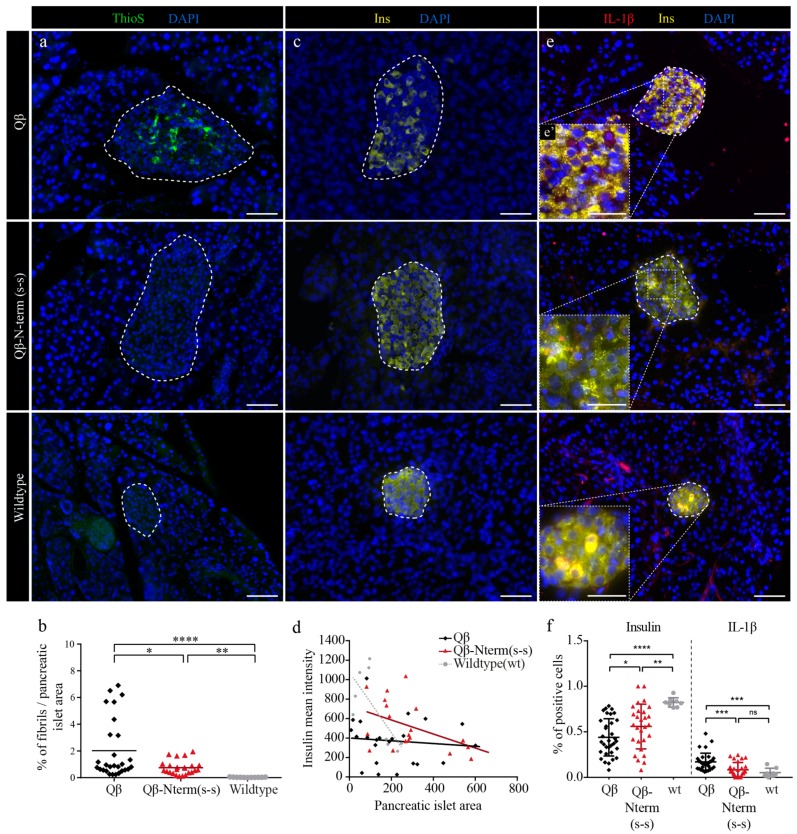
Immunization with the Qβ-Nterm(s-s) prevents fibrils depositions in pancreatic islets of hIAPP transgenic mice. (**a**,**c**,**e**) Paraffin-embedded pancreatic sections of 12-week-old Qβ (first row, Qβ, *n* = 32), Qβ-Nterm (s-s) (middle row, Qβ-Nterm (s-s), *n* = 28)-immunized homozygous male FVB/Ins-hIAPP transgenic mice and non-immunized wild-type FVB (last row, wildtype, *n* = 8) mice. (**a**) Fibril deposits were stained with ThioflavinS (ThioS, in green) and nuclei with DAPI (in blue). (**b**) Quantification of the relative amyloid/islet area in Qβ-Nterm (s-s) (in red), the Qβ control (in black)-immunized mice and the wild-type FVB control mice (in grey). (**c**) Insulin producing β-cells were stained with an anti-insulin antibody (in yellow) and nuclei with DAPI (in blue). (**d**) Correlation between the mean insulin intensity versus the pancreatic islet area of the Qβ control group (in black), the Qβ-Nterm (s-s)-immunized mice (in red) and the wild-type control (in grey). (**e**) Detection of IL-1β (in red), insulin (in yellow) in the pancreatic islets, and the nuclei with DAPI (in blue). (**e**) Magnification of the corresponding islet. (**f**) Quantifications of insulin-positive/total (left panel) and of the IL-1β-positive (right panel) pancreatic islet cells. Scale bars: 20 and 10 μm for the magnified pictures. Statistical test: Mann-Whitney test. Data are the means ± SEM. Ns: not significant, * *p* < 0.05, ** *p* < 0.01, *** *p* < 0.001, **** *p* < 0.0001. ThioS, ThioflavS; Ins, Insulin, IL-1β, Interleukin-1 β; wt, wild type.

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
