# Peer review of "Vaccination Against Amyloidogenic Aggregates in Pancreatic Islets Prevents Development of Type 2 Diabetes Mellitus"

_vaccines, 2020, doi:10.3390/vaccines8010116_

Round 1

Reviewer 1 Report

In my opinion, the authors submitted very interesting and well prepared paper. Its impact on the knowledge about diabetes mellitus is very strong. I want to emphasised high quality of the figures. In my opinion, this paper before acceptance in Vaccines needs only minor revision. My individual comments are listed below.

The aim of this study should be clearly presented at this end of the Introduction section.

5 – It should be “Type 2 diabetes mellitus”.

109 – It should be “sodium”.

124 – It should be “ribonuclease A”.

127 – It should be “horseradish peroxidase”.

128 – It should be “3,3’,5,5’-tetra….”

130 – It should be “ … and OD at wavelength at 450 was … “.

131 – It should be “the maximal OD at 450 nm”.

136 - It should be “… protein G Sepharose 4 Fast Flow …”

140 – Dialysis cut-off?

142 - It should be “hexafluoroisopropanol”.

153 – It should be “… Tween 20 … casein.”.

180 – It should be to 70% (v/v) ethanol”.

182 – It should be “xylol”.

186 – It should be “insulin”.

290 – It should be “Figure 2.”.

419 – Paper of Bram et al must be added to References.

Author Response

Please see the attachment "point-by-point comment reviewer 1_".

Reviewer 2 Report

This is an interesting manuscript describing vaccination against T2DM.

Could the authors clarify:

a. How were the numbers of animals for experiments determined i.e. how was the study powered?

b. Were the data normally distributed i.e. to justify parametric data presentation and analyses?

Author Response

Please see the attachment " point-by-point comment reviewer 2__".

Round 2

Reviewer 2 Report

The authors have satisfactorily responded to my comments.